# Optimized Acetic Acid Production by Mixed Culture of *Saccharomyces cerevisiae* TISTR 5279 and *Gluconobacter oxydans* TBRC 4013 for Mangosteen Vinegar Fermentation Using Taguchi Design and Its Physicochemical Properties

**DOI:** 10.3390/foods12173256

**Published:** 2023-08-29

**Authors:** Nisa Saelee, Ling-Zhi Cheong, Manat Chaijan

**Affiliations:** 1School of Agricultural Technology and Food Industry, Walailak University, Nakhon Si Thammarat 80160, Thailand; cmanat@wu.ac.th; 2School of Agriculture, Food and Ecosystem Sciences, Faculty of Science, University of Melbourne, Parkville, VIC 3010, Australia; lingzhi.cheong@unimelb.edu.au; 3Food Technology and Innovation Research Center of Excellence, Walailak University, Nakhon Si Thammarat 80160, Thailand

**Keywords:** vinegar, mangosteen, *Saccharomyces cerevisiae*, *Gluconobacter oxydans*, mixed culture

## Abstract

This research investigates the enhancement of acetic acid production in the mangosteen vinegar fermentation process through mixed-culture fermentation involving *S. cerevisiae* TISTR 5279 and *G. oxydans* TBRC 4013, alongside an analysis of the resulting mangosteen vinegar’s qualities and properties using Taguchi Experimental Design (TED). It focuses on key parameters, such as the juice concentration, inoculum ratio, and pasteurization conditions, to optimize acetic acid production. The findings highlight that the unpasteurized condition exerts the most significant influence on acetic acid production yield (*p* < 0.01), followed by the 3:1 inoculum ratio of *S. cerevisiae* TISTR 5279 to *G. oxydans* TBRC 4013 and a 10% mangosteen concentration. The achieved theoretical maximum yield of acetic acid on day 21 was 85.23 ± 0.30%, close to the predicted 85.33% (*p* > 0.05). Furthermore, the highest recorded acetic acid concentration reached 5.34 ± 0.92%. On day 14 of fermentation, the maximum productivity and yield were 3.81 ± 0.10 g/L/h and 0.54 ± 0.22 g/g, respectively. The resulting mangosteen vinegar exhibited elevated levels of total phenolic content (359.67 ± 47.26 mg GAE/100 mL), total flavonoid content (12.96 ± 0.65 mg CAE/100 mL), and anti-DPPH radical activity (17.67 ± 0.22%), suggesting potential health benefits. Beyond these chemical aspects, the mangosteen vinegar displayed distinct physical and chemical characteristics from the original mangosteen juice, possibly conferring additional health advantages. These findings are promising for industrial vinegar fermentation models and propose the potential use of the product as a valuable dietary supplement.

## 1. Introduction

Mangosteen (*Garcinia mangostana* L.) from the Clusiaceae family originated in Southeast Asia and has spread to tropical regions like Thailand, Indonesia, Malaysia, southern India, northern Australia, Central America, Brazil, and Hawaii, earning it the title “Queen of Fruits” [1]. The fruit consists of two key parts, the succulent pulp and protective pericarps, with both seeds and pericarps being utilized extensively in traditional therapeutic practices in the regions [2]. Packed with bioactive compounds such as phenolic acids, xanthones, tannins, flavonoids, alpha- and beta-mangostins, and anthocyanidins [2,3,4,5,6,7,8,9,10], mangosteen offers a range of health benefits, including anti-inflammatory effects [4], antioxidative properties [6,8], anti-skin cancer potential [7], anti-acne effects [9], anti-obesity effects [2], and the inhibition of tumor growth [5]. While these compounds are largely concentrated in the peel and rind, the major phenolic acid in mangosteens aril is para-hydroxybenzoic acid [10]. Ongoing research specifically focuses on enhancing acetic acid production during the mangosteen vinegar fermentation process.

Vinegar, an acetic acid-containing product, has long served as a versatile flavor enhancer in foods and functional beverages. Its fermentation process imbues it with acetic acid, a bioactive compound known for its medicinal, antioxidative, antibacterial, anti-infective, anticancer, and hyperglycemic properties. Moreover, vinegar’s polyphenols are believed to confer protection against conditions such as cancer, lipid peroxidation, hypertension, hyperlipidemia, inflammation, and DNA (deoxyribonucleic acid) damage [11]. Diverse raw materials, including black rosehip [12], balsamic vinegar [13], *Hericium erinaceus* [14], citrus [11,15], lemon juice [16], strawberry [17], rice [18], cereal [19], banana peel [20], baby corn silk [21], and *Cordyceps militaris* waste [22], are employed in vinegar production. The characteristics of different vinegar types are shaped by factors such as raw materials, fermentation methods, processes, and aging techniques [23,24]. Vinegar fermentation is influenced by variables such as yeast strain, acetic acid bacteria (AAB), pH, temperature, oxygen, and fermentation methods. Notably, yeast strains and AAB [25] hold significance, as they yield distinct volatile compounds and alcohol levels, resulting in varying vinegar profiles in terms of aroma, alcohol content, and acetic acid concentration [25]. Starter cultures are commonly applied to enhance the flavor of fermented foods [26]. 

In the process of vinegar fermentation, acetic acid and ethanol emerge as pivotal molecules. Both traditional and commercial vinegar production entail two sequential fermentation phases: firstly, yeast initiates an aerobic ethanol fermentation, followed by the conversion of ethanol to acetic acid by acetic acid bacteria (AAB). Certain bacteria have demonstrated their ability to partake in vinegar fermentation, with notable yeast strains like *S. cerevisiae* [27,28,29,30], and significant AABs such as *Acetobacter* and *Gluconobacter*, the latter being crucial for converting ethanol into substantial amounts of acetic acid [31]. Acetobacter effectively transforms ethanol into acetic acid and subsequently fully oxidizes it into water and carbon dioxide [32]. Conversely, *Gluconobacter* bacteria exhibit a preference for sugar as a carbon source, serving as a vital component in the industrial production of 1,3-dihydroxyacetone, gluconic acid, vitamin C, and vinegar. Yet, *G. oxydans* NL71 displays the capacity to ferment xylose into xylonic acid, thereby offering a pathway for highly efficient fermentation [33].

Vinegar production involves a variety of fermentation methods, encompassing conventional techniques [34,35], submerged fermentation (SMF) [17], and solid-state fermentation (SSF) [19]. SMF boasts several advantages, such as higher yields and shorter fermentation times; however, the resulting vinegar is comparatively less potent than that from SSF [11]. Traditional vinegar fermentation proves suitable only for small-scale production and specific juices, occurring in non-sterile environments with assorted unidentified strains. These traditional bioprocesses present difficulties in standardization and often lack the required efficiency when upscaled [35], given the distinct growth preferences of various microbes that can lead to competitive dynamics. While mixed-culture fermentation is notably efficient and straightforward, with minimal capital requirements, its management on an industrial scale remains challenging [28].

In this research, mixed-culture fermentation involving *S. cerevisiae* TISTR 5279 and *G. oxydans* TBRC 4013 is subjected to statistical optimization to enhance acetic acid production in the process of mangosteen vinegar fermentation. Following this, an investigation into the qualities and properties of the resulting mangosteen vinegar is undertaken.

## 2. Materials and Methods

### 2.1. Preparation of Raw Materials and Starter Cultures 

The experimental raw material, mangosteen fruit, was procured from a local farm in Nakhon Si Thammarat, Thailand. Ripe fruits in stage 5 of maturity, indicated by a dark blackish purple peel color (as illustrated in Figure 1), devoid of defects, were meticulously chosen. After cleaning and peeling, the soft white pulp along with the seeds were gathered and stored at −20 °C for the experimentation. The evaluation encompassed a range of parameters, including total soluble solids (TSS, °Bx), total titratable acidity (TTA), sugar content, total phenolic content (TPC), total flavonoid content (TFC), antiradical activity, turbidity, color, and mineral composition of the juice extracted from the soft white pulp. The procedures specified in Section 2.4 were employed to determine these attributes.

*Saccharomyces cerevisiae* TISTR 5279 was purchased from the Thailand Institute of Scientific and Technological Research (TISTR), while *Gluconobacter oxydans* TBRC 4013 was purchased from the Thailand Bioresource Research Center (TBRC).

A single loop of S. cerevisiae TISTR 5279 maintained on potato dextrose agar (PDA) (HimediaTM, India) slant at 4 °C, and the strain was streaked onto PDA agar plates and incubated at 30 °C for 48 h. A single colony was then transferred to a 500 mL flask containing 100 mL of yeast malt (YM) broth comprising 0.3% yeast extract, 0.5% peptone, 0.3% malt extract, and 1% glucose. The culture was incubated at 30 °C for 48 h. The inoculum’s optical density at 600 nm was adjusted to 3.0 before it was utilized as the seed inoculum.

The stock culture of *G. oxydans* TBRC 4013, preserved on a glucose/ethanol/calcium carbonate agar (GECA, containing 25 g/L glucose, 5 mL ethanol, 3 g/L yeast extract, 5 g/L peptone, 7 g/L CaCO_3_, and 12 g/L agar) slant at 4 °C, was streaked onto GECA agar plates and then incubated at 30 °C for a duration of 48 h. Subsequently, a single colony was transferred to a 250 mL flask containing 100 mL of glucose/yeast extract/peptone (GYP) medium, composed of 20 g/L glucose, 12 g/L peptone, and 3 g/L yeast extract. The culture was cultivated at 30 °C for 48 h to prepare the seed. The starter culture was adjusted to an absorbance of 3.0 at 600 nm before utilization. 

### 2.2. Mixed-Culture Optimization Parameters 

Mangosteen pulp was manually squeezed, filtered through a cloth filter, and separated from the seed. To be employed in this investigation, the TSS concentration of the mangosteen juice was diluted to 5 and 10 °Bx using sterilized deionized water (DI). The initial pH of the mangosteen juice was adjusted to 6.6 with 1 M NaOH. 

Following our preliminary investigation, the mixed culture involving a single-stage fermentation process was selected as the optimal condition for acetic acid production. After thermal pasteurization at 65 °C for 20 min, the mangosteen juice was inoculated with an inoculum comprised of a mixture of *S. cerevisiae* TISTR 5279 and *G. oxydans* TBRC 4013 seed. A working volume of 100 mL was established in a 500 mL flask, into which a 1% *v*/*v* seed culture, adjusted to a specific absorbance, was introduced.

Under these circumstances, 100 mL of medium working volume in a 500 mL flask was inoculated with 1% *v*/*v* adjusted absorbance seed culture. Co-culture fermentation took place for 14 days at a temperature of 30 °C and a shaking speed of 150 rpm. Before further analysis, the fermentation broth was collected through centrifugation at 6000× *g*, and the supernatant was stored at −20 °C. The contents of acetic acid, sugars, and ethanol within the supernatant were evaluated utilizing high-pressure liquid chromatography (HPLC). The computation of the theoretical acetic acid yield was facilitated through the utilization of the following equation:(1)$$Acetic\;acid\;yield\left(\%Theoritical\right)=\frac{Cac\times 100}{0.70175\times Csu}$$

Here, the variable C*ac* represents the acetic acid concentration, while C*su* denotes the total initial sugar concentration derived from sucrose. The factor 0.70175 is employed as the conversion coefficient, facilitating the transformation from sucrose to acetic acid.

The acetic acid yield (Y*ac/s*) (g/g) was calculated as follows:(2)$$Yac/s=\frac{Total\;acetic\;acid\;produced\;(g)}{Total\;sugar\;consumed\;(g)}$$

The acetic acid productivity (Q*ac*) (g/L/h) was calculated as follows: (3)$$Qac=\frac{Acetic\;acid\;concentration\;(g/L)}{Fermentation\;time\;(h)}$$

### 2.3. Taguchi Design of Experiment 

To determine the factors affecting acetic acid production during mangosteen vinegar production, the design of experiment (DOE) employing Taguchi Experimental Design (TED) was created using Minitab Statistical Analysis Software (Minitab 18, Minitab Inc., State College, PA, USA) to clarify the important factors and their significance on the response. Three factors with 24 levels were set for the experiments (Table 1). There were three parameter factors: (A) the mangosteen juice concentration (°Bx), with two levels of 5 and 10 °Bx; (B) the inoculum ratios between *S. cerevisiae* TISTR 5279 and *G. oxydans* TBRC 4013, with four levels of 0:4, 2:2, 1:3, 3:1; and (C) the pasteurization conditions, with two levels of pasteurized and unpasteurized conditions, where pasteurization of the medium was performed through autoclaving at 65 °C for 20 min.

L16 (4^1^ 2^2^), a mixed 2–4 level design of an orthogonal array was selected for 16 experiment tests. Table 2 presents an overview of the design of the study. The acetic acid yield responses (%theoretical) were evaluated. The analysis of variance (ANOVA) was used to discuss the relative effect and determine the highest effect factors at *p*-values < 0.05. Applying the principle “bigger is better” involves using ANOVA for signal-to-noise (S/N) ratios. The following equation was used to examine the display response, which was expressed as acetic acid yield (%theoretical) for means or an acetic acid yield (%theoretical) for S/N ratios:(4)$$S/N=-10\times log(\sum \frac{\frac{1}{Y^{2}}}{n})$$

In this context, Y signifies the count of responses corresponding to a particular combination of factor levels, while n represents the total count of responses within that same factor level combination.

**Table 2 foods-12-03256-t002:** L16 (4^1^ 2^2^) orthogonal array using Taguchi Design.

Run	A	B	C	Acetic Acid Yield (%Theoretical)
1	1	1	1	7.1
2	1	1	1	5.7
3	2	1	2	62.2
4	2	1	2	53.6
5	1	2	1	4.3
6	1	2	1	2.9
7	2	2	2	76.3
8	2	2	2	99.6
9	1	3	2	83.7
10	1	3	2	84.6
11	2	3	1	9.6
12	2	3	1	9.0
13	1	4	2	90.5
14	1	4	2	86.8
15	2	4	1	8.3
16	2	4	1	4.3

Based on Taguchi’s predictive approach, the acetic acid production within the mangosteen vinegar fermentation was verified in a 500 mL flask, employing a composition of 10% mangosteen juice and 1% inoculum in a 3:1 ratio (*S. cerevisiae* TISTR 5279: *G. oxydans* TBRC 4013), without subjecting the medium to pasteurization. The flask underwent a 28 days fermentation period at 30 °C and 200 rpm. At two-day intervals, samples were collected for subsequent analyses. Acetic acid, ethanol, sugars, and the viable growth of *S. cerevisiae* TISTR 5279 and *G. oxydans* TBRC 4013 were presented as the mean ± standard deviation (SD) based on duplicated data.

### 2.4. Analytical Procedures

The total soluble solid (TSS), reported as °Bx, was measured using a hand refractometer (ATAGO, Tokyo, Japan). 

The pH value was determined using a pH meter (WTW series: InoLab pH730, Weilheim, Germany). The titratable acid was tested using the titration method with 0.01 N NaOH as a titrant and phenolphthalein as an indicator. 

The total acidity (%) was calculated as total acidity (%) = N NaOH × mL Titrant × 0.006 × 100)/mL sample. 

The turbidity value was tested using a turbidimeter (Model 2100N IS, HACH, Loveland, CO, USA) with a standard 20, 200, 1000, and 2000 NTU, and the final turbidity value was reported as an actual NTU calculated as a measured result × dilution factor. 

The color value of the mangosteen vinegar sample was measured using Colorimeter (HunterLab, Colorflex EZ 45-0, Reston, WV, USA). The color values were expressed as *L** (lightness), *a** (redness/greenness), and *b** (yellowness/blueness).

The viable growth of *S. cerevisiae* TISTR 5279 and *G. oxydans* TBRC 4013 were examined using the spread plate technique on the PDA agar plate and GECA agar plate, respectively. The total viable cells were measured after incubation at 30 °C for 48 h. The number of colony forming units per mL (CFU/mL) was calculated according to the number of colonies on the plates (30–300 colonies) × the dilution factor.

The sugar contents of raw material and mangosteen vinegar products were determined using the HPLC system (Waters, Milford, MA, USA) using Animex HPX-87P column, 300 × 7.8 mm (Biorad, Hercules, CA, USA), as previously described [36]. The mobile phase, DI, was performed at 0.6 mL/min, in an 80 °C column oven with a 50 °C refractive index (RI) detector. The samples were diluted with DI to the proper dilution and filtered using a 0.45 μm filter. Ten µL of the filtered sample was injected. The external standards of 1%, sucrose, D (−)-fructose, and D-(+)-glucose anhydrous were used. A linear relationship was created for quantification.

The ethanol, acetic acid, and sugar contents during fermentation were analyzed using the HPLC method (Waters, Milford, MA, USA) with an Aminex HPX-87H column (300 × 7.8 mm, Bio-Rad, CA, USA) using 0.004 M H_2_SO_4_ at a flow rate of 0.6 mL/min at 55 °C for 30 min. For the measurement of ethanol and sugars, a RI detector at 50 °C was used with an ethanol and sugars standard. The amount of acetic acid was measured with a UV detector at 200–210 nm, and acetic acid was used as a reference standard. Individual sugars, acetic acid, and ethanol were measured using calibration curves derived from standard substances.

An inductively coupled plasma optical emission spectrometer (ICP-OES, Varian 730-ES, Mulgrave, Australia) was used to analyze the mineral content in mangosteen juice and vinegar using a TraceCERT^®^, with 33 elements (Sigma-Aldrich, Burlington, MA, USA) as a standard. Ten mL of the sample was combined with 2% nitric acid to the desired concentration. Before analysis, the samples were centrifuged or filtered through Whatman No. 4 to remove any precipitate that had formed. The standards, calcium, phosphorous, sodium, potassium, magnesium, sulfur, copper, zinc, aluminum, manganese, nickel, zinc, and boron were tested at three different concentrations (0.5–25 mg/L). The quantitative levels of each trace were determined using their standard calibration curve [36].

Total phenolics content (TPC) was determined using the Folin-Ciocalteau method, according to Wang et al. [37]. The sample was filtered through a 0.45 µm membrane filter. A 2 mL clear centrifuged supernatant of the sample, 2.5 mL of 10% Folin–Ciocalteu reagent, and 2 mL of 7% Na_2_CO_3_ were added, thoroughly mixed, and incubated at room temperature (27–29 °C) for 1 h. The absorbance at 765 nm was measured using gallic acid (Sigma Aldrich, Burlington, MA, USA) as a linear calibration standard. The total phenolics content was expressed in mg of gallic acid equivalents (GAE) per 100 mL.

The total flavonoid content (TFC) was determined using the method described by Wang et al. [37]. After filtering the sample through a 0.45 µm membrane filter, 1 mL of the sample was added to with 4 mL of DI, 0.3 mL of 5% NaNO_2_, 0.3 mL of 10% AlCl_3_, and 2 mL of 1 M NaOH, the mixtures were thoroughly vortexed, and the absorbance was measured at 510 nm using catechin (Sigma Aldrich, USA) as a linear calibration standard. The total flavonoid content was expressed in mg of catechin equivalents (CAE) per 100 mL.

The antiradical activity was determined using the modified method of Wang et al. [37]. The sample was filtered through a 0.45 µm membrane filter, and a 0.1 mL aliquot was pipetted, added with 5.0 mL, 0.1 mM 2,2-Diphenyl-1-picrylhydrazylradical (DPPH), thoroughly mixed, and kept in the darkness at room temperature for 30 min. The absorbance at 517 nm was measured. The antiradical activity (%) was calculated as (Ac − As)/Ac) × 100, where Ac is the absorbance of the control reaction using 0.1 mL methanol as a sample, and As is the absorbance of the sample reaction.

### 2.5. Statistical Analysis

All the data were analyzed using one-way ANOVA with Minitab 18.0 (Minitab Inc., USA). The means were compared using Fisher pairwise comparisons. All analyses were carried out in triplicate. The data are presented as Means ± SD.

## 3. Results and Discussion

### 3.1. Mangosteen Juice Properties

The consumable portion of the mangosteen fruit, encompassing the white pulp along with the seed, constituted 20.61% *w*/*w*, while the mangosteen juice segment contained 10.88% *w*/*w* (as illustrated in Figure 1). The peel’s proportion amounted to 62.2%. Some research suggests that the pericarp contributes around 70% to the mangosteen fruit composition [38], and roughly 40% belongs to the edible section [39]. The chemical composition of juice is influenced by the processes of fruit handling, processing, and storage [40]. With the intention of optimizing vinegar fermentation, a characterization of the chemical constituents present in fresh mangosteen juice extract was conducted. According to Table 3 in this study, sucrose emerged as the primary sugar in mangosteen juice, accounting for 14.07 ± 0.26% *w*/*v*, while reducing sugars, including glucose and fructose, were identified at 2.06 ± 0.12% *w*/*v* and 1.78 ± 0.08% *w*/*v*, respectively.

The mangosteen juice extracted from the white soft pulp exhibited a TSS of 17.18 ± 0.77 °Bx, accompanied by a corresponding total sugar content comprising sucrose, glucose, and fructose, which measured at 17.91 ± 0.43% (refer to Table 3). This surpassed the TSS values reported for pineapple and citrus juices, noted as 12.5–12.8 and 11.6–11.2 °Bx, respectively [41,42]. The elevation in TSS aligns with fruit ripeness, attributable to starch conversion into sugars [43]. However, it is important to note that TSS and sugar content can vary between different fruits. The freshly squeezed mangosteen juice possessed a titratable acidity of 1.26 ± 0.20% and an acidic pH of 3.66 ± 0.77, closely resembling the pH range observed in pineapples, which falls between 3.56 and 3.69 [41].

Among the essential minerals identified in the fresh mangosteen pulp extract, the sodium content measured 2.69 ± 0.74 mg/100 g, potassium accounted for 38.38 ± 1.62 mg/100 g, phosphorous was noted at 13.01 ± 0.23 mg/100 g, magnesium at 17.91 ± 0.39 mg/100 g, calcium at 7.01 ± 0.72 mg/100 g, and sulfur at 4.54 ± 1.63 mg/100 g (as outlined in Table 3). The concentrations of aluminum, manganese, iron, nickel, copper, boron, and zinc, however, could not be ascertained. Compared with canned mangosteen (syrup pack), the levels of phosphorous and magnesium were notably higher in the fresh pulp, registering at 9.21 mg/100 g and 13 mg/100 g, respectively [44]. On the other hand, potassium and calcium exhibited lower concentrations, measuring 48 mg/100 g and 12 mg/100 g, respectively.

Mangosteen juice displayed TPC and TFC values of 427.39 ± 148.41 mgGAE/100 g and 67.80 ± 15.43 mg/100 g, respectively, as outlined in Table 3. The DPPH radical scavenging activity of mangosteen juice was found to be 55.29 ± 28.08%. The evaluation of the compounds’ potential to function as free-radical scavengers involves the utilization of the DPPH free radical, which underscores the variations in phenolic compounds between mangosteen pulp and peel [3]. Visualized in Figure 1, the mangosteen juice possesses a deep pink-purple turbidity with *L** (10.84 ± 4.65), *a** (0.89 ± 0.34), *b** (1.00 ± 0.43), and turbidity values amounting to 493.07 ± 84.25 NTU.

### 3.2. Optimization of Mangosteen Vinegar Production by Taguchi Design

For 16 design runs, the response values expressed as acetic acid yield (%theoretical) were given in Table 2. Table 4 displays the results of the ANOVA for the lactic acid yield. The estimated linear model coefficients for the effects of S/N ratios on acetic acid production were examined and fitted to a linear model, as follows:YD14 = 42.14 + 3.56A1 − 3.56A2 − 7.84B1 − 2.08B2 + 4.59B3 + 5.33B4 − 5.74C1 + 35.74C2(5)
YD21 = 58.31 + 2.92A1 − 2.92A2 − 21.69B1 − 18.42B2 + 15.15B3 + 24.96 B4 − 4.98C1 + 4.98C2(6)
where YD14 is the lactic acid yield (%theoretical) at day 14 of fermentation, and YD21 is the lactic acid yield (%theoretical) at day 21 of fermentation. A, B, and C are the coded factors for mangosteen juice concentration (°Bx), inoculum seed ratios, and pasteurization conditions, respectively. Numbers 1, 2, 3, and 4 are the coded levels of each parameter as indicated in Table 1.

The ANOVA results indicated that all the parameter tests, encompassing the mangosteen juice concentration, inoculum seed ratios, and pasteurization conditions, exerted a noteworthy impact (*p* < 0.01) on the acetic acid yield during the 14-day fermentation process of mangosteen vinegar. Conversely, by day 21, only the inoculum seed ratios demonstrated a substantial influence (*p* < 0.01) on the resultant acetic acid yield.

The linear regression equation model estimated during days 14 and 21 of fermentation displayed high R^2^ values of 99.17% and 99.44%, along with adjusted R^2^ values of 97.09% and 80.55%, respectively. This observation suggested a significant influence (*p* < 0.01) of the optimization parameters on the lactic acid yield within the mixed culture fermentation of mangosteen vinegar involving *S. cerevisiae* TISTR 5279 and *G. oxydans* TBRC 4013.

The acetic acid yield exhibited an influence sequence of C > B > A and B > C > A, at day 14 and 21 of fermentation, respectively, as presented in Table 5. Consequently, the ratio between the mixed seed and the utilized inoculum emerges as the pivotal variable in mangosteen vinegar production. Notably, a greater delta value for a parameter signifies a more pronounced impact on the response. The interaction plots for each parameter are depicted in Figure 2, revealing a notable aspect: the non-parallel nature of the lines within this plot indicates a substantial relationship between the factors and the corresponding response values.

The optimal prediction for the acetic acid yield was achieved by utilizing 10% mangosteen juice, maintaining an inoculum seed ratio of *S. cerevisiae* TISTR 5279 to *G. oxydans* TBRC 4013 at 3:1, and employing unpasteurized conditions. The anticipated acetic acid yield was projected to reach 79.66% and 85.33% of the theoretical maximum at 14 and 21 days of fermentation, respectively, as detailed in Table 6.

The absence of pasteurization potentially fostered the development of a diverse microbial community, leading to vinegar characterized by unique quality and substantial acetic acid yield [45]. Pasteurized conditions, on the other hand, tend to result in sluggish fermentation due to the lag phase experienced by microorganisms while adapting to new media [46]. It is noteworthy that most vinegar production refrains from employing starter cultures due to its limited commercial value and fermentation complexities [47]. Nevertheless, the flavor compounds generated through spontaneous fermentation are closely linked to the diversity of the microorganisms involved [48].

### 3.3. Acetic Acid Production under Optimized Conditions by the Mixed S. cerevisiae TISTR 5279 and G. oxydans TBRC 4013 

Pursuant to the anticipated conditions, a non-sterile mixed-culture fermentation was undertaken, with the results elucidated in Figure 3 and Table 7. The fermentation of mangosteen vinegar was executed through a blend of *S. cerevisiae* TISTR 5279 and *G. oxydans* TBRC 4013. Within the optimized fermentation framework, the initial sugar levels reached 11.04 ± 4.03%, which rapidly diminished and were completely consumed during the 2–4-day window of fermentation, aligning with ethanol and acetic acid production. *S. cerevisiae* TISTR 5279 demonstrated its highest cell density at 1.34 × 10^9^ CFU/mL (initially 7.5 × 10^7^ CFU/mL), prompting the swift generation of ethanol, peaking at 3.95% on day 4. However, the ethanol concentration exhibited variations between 0.73% and 3.95%, being wholly utilized by day 21. As a consequence, the acetic acid content commenced its ascent on day 2, concomitant with the elevation of *G. oxydans* TBRC 4013 to around 1.30 × 10^9^ CFU/mL from initially 5.5 x 10^7^ by day 8. The high levels of acetic acid materialized at 5.33 ± 0.14 and 5.34 ± 0.92% with 75.94 ± 1.99 and 85.23 ± 30%theoretical on days 14 and 21, respectively. *G. oxydans* TBRC 4013 persisted in converting ethanol to acetic acid. The noted report accentuates the varying growth abilities of AAB, often challenging the initiation of acidification [47]. This increase in acetic acid, coupled with the reduction in reducing sugars, was linked to the prevailing growth of *Lactobacillus* and *Acetobacter* [49].

The theoretical acetic acid yields were 75.94 ± 1.99 and 85.23 ± 30%theoretical on days 14 and 21 of fermentation, respectively, close to the predicted values of 79.65% and 85.33% on the respective days (*p* > 0.05). The productivity and yield reached 3.81 ± 0.10 g/L/h and 0.54 ± 0.22 g/g, and 2.54 ± 0.44 g/L/h and 0.51 ± 0.10 g/g on days 14 and 21, respectively, as depicted in Figure 3. The most pronounced effective inoculum ratio determined in this study was a 3:1 ratio between *S. cerevisiae* TISTR 5279 and *G. oxydans* TBRC 4013, realized through a single-stage mixed culture fermentation. The higher acid content may be attributed to the shared optimal temperature of 30 °C for *S. cerevisiae* (TISTR 5279) and *G. oxydans* TBRC 4013. The co-culture of *G. oxydans*, a robust aerobe, and *S. cerevisiae*, a facultative anaerobe, within a shake flask rotating at 200 rpm likely satisfied the organisms’ oxygen requisites. While the optimal pH for AAB growth ranges from 5 to 6.5, they can thrive in the lower pH range of 3 to 4 [31], posing challenges for sampling during shake flask fermentation. It is worth noting that the acetic acid content was lower than the reported value of 89.3 g/L in the mixed culture of *S. cerevisiae* and *Acetobacter pasteurianus* (ranging from 75.94 ± 1.99 g/L to 76.01 ± 13.05 g/L) [28]. In persimmon vinegar, the ethanol and acidity content peaked at 5.18 ± 1.03% (*v*/*v*) and 46.30 ± 8.29 g/L, respectively [48]. Ethanol in Zhejiang Rosy Vinegar, fermented by *Lactobacillus* and *Acetobacter*, swiftly escalated to 6.63% by volume, accompanied by 4.85 g/100 mL titratable acid [49].

### 3.4. Physical and Chemical Properties of Mangosteen Vinegar

Several physical attributes can influence the sensory evaluation of the mangosteen vinegar product. The juice’s initial pH decreased from 6.6 ± 0.00 to 2.95 ± 0.06 and 3.01 ± 0.0 (*p* < 0.05), while its acidity (%) increased to a peak of 4.94 ± 1.11% at 21 days, higher than starting value of 0.60 ± 0.00% (*p* < 0.01) (as indicated in Table 8). The mangosteen vinegar product revealed reducing turbidity in comparison to the outset of fermentation (271.00 ± 1.41 NTU) (*p* < 0.05). A lower turbidity of 19.11 ± 0.04 and 12.07 ± 0.99 NTU on day 14 and day 21, respectively, implied the removal of fewer suspended solids during vinegar production, a preference for consumers. These observations underscore significant physical and chemical transformations aimed at augmenting clarification throughout the mangosteen fermentation process.

The TPC of mangosteen vinegar exhibited a decline from its initial value of 440.25 ± 2.59 mg GAE/100 mL to 359.67 ± 47.26 and 200.51 ± 7.08 mg GAE/100 mL on days 14 and 21, respectively, corresponding to the diminishing TFC (20.38 ± 5.08, 12.96 ± 0.65, 11.63 ± 0.00 mg/100 mL) and DPPH radical scavenging activity (18.75 ± 7.40, 17.67 ± 0.22, 15.17 ± 2.47%) at the initial, 14 and 21 days, respectively. Nevertheless, TFC and TPC do not significantly change during increased time of fermentation (*p* > 0.05). Polyphenols, which are abundant in fruits and vegetables and known for their potent antioxidant activity and therapeutic attributes [50], contribute to these observed changes. In contrast to flesh-derived mangosteen vinegar, this study’s TPC and TFC content varied, offering 541.40 ± 18.80 mg GAE/g extracted, and 0.21 ± 0.01 mg QE/g extracted, respectively, utilizing a high-pressure autoclave procedure [51].

Color represents a pivotal quality aspect in food evaluation, with polyphenols, highly concentrated in fruits and vegetables, bearing both remarkable antioxidant prowess and medicinal potential [50]. The purple pericarp of the mangosteen, enriched with anthocyanins, a natural red pigment, not only imparts an appealing hue but also bestows significant anti-inflammatory and antioxidant attributes [38]. Recorded in Table 8 are the color metrics of mangosteen vinegar pre- and post-fermentation. A slight pinkish tint was observed in the mangosteen juice, possibly originating from the pericarp’s proximity to the white pulp during peeling. The *L** value exhibited no reduction post-fermentation, measuring 3.62 ± 0.02 and 3.93 ± 0.66 at 14 and 21 days, respectively, compared to the initial 4.11 ± 0.01 (*p* > 0.05)., reflecting the transparency of the vinegar. Notably, the *a** values indicated no change in redness, while the *b** values indicated a heightened yellowness in the vinegar (*p* > 0.05). Color changes can result from various reactions such as acid oxidation, heating, and enzymatic browning, leading to anthocyanin degradation and the emergence of browning pigments, limiting the utilization of red hues [38]. The elevated *L** value of the original white mangosteen pulp extract denotes darkening, a phenomenon consistent with the transformation seen in mangosteen vinegar’s color before and after fermentation, illustrated in Figure 1. Moreover, significant elements (as presented in Table 9), including sodium (11.60 ± 0.05 mg/100 mL), potassium (234.45 ± 1.06 mg/100 mL), phosphorous (55.43 ± 3.33 mg/100 mL), magnesium (96.02 ± 6.19 mg/100 mL), calcium (34.84 ± 1.81 mg/100 mL), and sulfur (76.54 ± 41.95 mg/100 mL) were evident in mangosteen vinegar, offering potential health benefits.

## 4. Conclusions

In essence, vinegar production typically involves dual-phase fermentation: alcoholic fermentation followed by acetic acid fermentation. However, this study effectively optimized a mixed-culture fermentation of *S. cerevisiae* TISTR 5279 and *G. oxydans* TBRC 4013 within a single-stage process. Through statistically optimized SMF, heightened acetic acid concentrations were attained using unpasteurized mangosteen juice at 10 °Bx with a 3:1 inoculum ratio (*S. cerevisiae* TISTR 5279 and *G. oxydans* TBRC 4013). Furthermore, the mangosteen vinegar exhibited physicochemical traits such as TPC, TFC, DPPH, and mineral content that could contribute to its flavor profile. This underscores the potential of mangosteen fruit to be used as a valuable substrate for vinegar production. Nonetheless, to enhance the flavor nuances of mangosteen vinegar, an exploration of flavor- and aroma-impacting volatile and bioactive compounds is recommended. Furthermore, an extensive-scale mangosteen vinegar fermentation warrants investigation.

## Figures and Tables

**Figure 1 foods-12-03256-f001:**
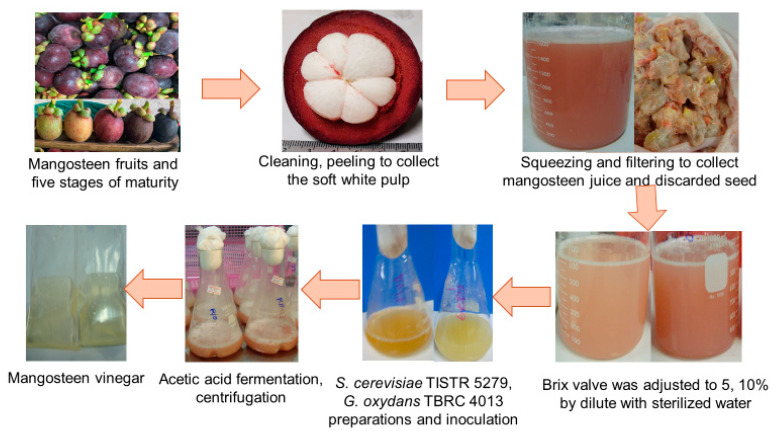
Mangosteen vinegar production process.

**Figure 2 foods-12-03256-f002:**
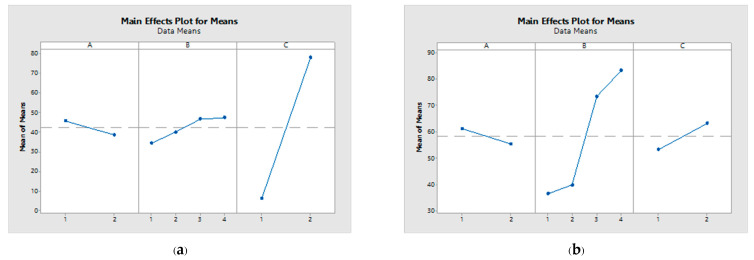
Main effect plot for means at 14 d (**a**) and 21 d (**b**) of fermentation.

**Figure 3 foods-12-03256-f003:**
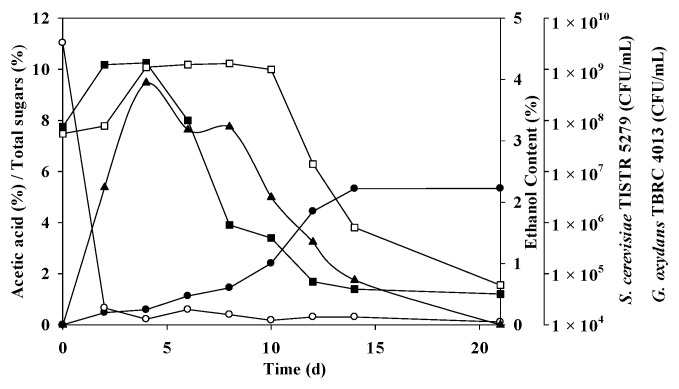
Mangosteen vinegar fermentation with mixed cultures of *S. cerevisiae* TISTR 5279 and *G. oxydans* TRBC 4013. Acetic acid (●), ethanol (▲), residual sugars (○), *G. oxydans* TRBC 4013 (□), and *S. cerevisiae* TISTR 5279 (■) are indicated.

**Table 1 foods-12-03256-t001:** Selected parameters with three factors and 2–4 levels.

Codes	Factors	Level 1	Level 2	Level 3	Level 4
A	Mangosteen juice concentration (°Bx)	5	10	-	-
B	Inoculum ratios between *S. cerevisiae* TISTR 5279 and *G. oxydans* TBRC 4013	0:4	2:2	1:3	3:1
C	Pasteurization conditions	Pasteurized	Unpasteurized	-	-

**Table 3 foods-12-03256-t003:** Chemical properties of the white pulp mangosteen juice.

Characteristics			
pH	3.66 ± 0.77	Minerals (mg/100 g)	
Sugars (g/L)	17.91 ± 0.43	Sodium	2.69 ± 0.74
Sucrose	14.07 ± 0.26	Potassium	38.38 ± 1.62
Glucose	2.06 ± 0.12	Phosphorous	13.01 ± 0.23
Fructose	1.78 ± 0.08	Magnesium	17.91 ± 0. 39
Titratable acids (%)	1.29 ± 0.20	Calcium	7.01 ± 0. 72
TSS (°Bx)	17.18 ± 0.77	Sulphur	4.54 ± 1.63
TFC (mg/100 g)	67.80 ± 15.43	Aluminium	nd
TPC (mgGAE/100 g)	427.39 ± 148.41	Manganese	nd
DPPH radical inhibition (%)	55.29 ± 28.08	Iron	nd
Turbidity	493.07 ± 84.25	Nickle	nd
Color values		Copper	nd
*L**	10.84 ± 4.65	Zinc	nd
*a**	−0.89 ± 0.34		
*b**	1.00 ± 0.43		

All data was performed in triplicate and presented as means ± SD. nd = not detected.

**Table 4 foods-12-03256-t004:** ANOVA results of lactic acid yield for mangosteen vinegar with mixed culture fermentation of *S. cerevisiae* TISTR 5279 and *G. oxydans* TBRC 4013 using Taguchi design.

ANOVA Analysis Results												
	Day 14						Day 21					
**Source**	DF	Seq SS	Adj SS	Adj MS	F	P	DF	Seq SS	Adj SS	Adj MS	F	P
**A**	1	101.2	101.2	101.2	2.29	0.269	1	68.31	68.31	68.31	0.65	0.506
**B**	3	230.6	230.6	76.9	1.74	0.385	3	3324.45	3324.45	1108.15	10.49	0.088
**C**	1	10,221.5	10,221.5	10,221.5	231.26	0.004	1	198.79	198.79	198.79	1.88	0.304
**Residual Error**	2	88.4	88.4	44.2			2	211.35	211.35	105.67		
**Total**	7	10,641.6					7	3802.9				
	R^2^ = 99.17, adj.R^2^ = 97.09						R^2^ = 99.44, adj.R^2^ = 80.55					

The coded factors, A = Mangosteen juice concentrations, B = Inoculum seed ratios, and C = Pasteurization conditions. DF = Degrees of Freedom, Seq SS = Sequential sums of squares, Adj SS = Adjusted sum of squares, Adj MS = Adjusted mean squares, F = F-value (The test statistic), P = P-Value (Probability).

**Table 5 foods-12-03256-t005:** Response table for means using Taguchi analysis for optimum acetic acid production yield with the mixed *S. cerevisiae* TISTR 5279 and *G. oxydans* TBRC 4013 during mangosteen vinegar fermentation.

Response Table for Means						
	Day 14			Day 21		
**Level**	A	B	C	A	B	C
**1**	45.69	34.29	6.392	61.23	36.62	53.33
**2**	38.58	40.06	77.88	55.39	39.89	63.3
**3**		46.72			73.47	
**4**		47.47			83.27	
**Delta**	7.11	13.18	71.49	5.84	46.64	9.97
**Rank**	3	2	1	3	1	2

**Table 6 foods-12-03256-t006:** Optimum prediction results using Taguchi analysis for acetic acid production yield with the mixed *S. cerevisiae* TISTR 5279 and *G. oxydans* TRBC 4013 during mangosteen vinegar fermentation.

Fermentation Time (day)	Prediction			Settings		
	S/N Ratio	Mean	StDev	A	B	C
14	38.53	79.65	2.32	2	4	2
21	38.39	85.33	13.13	2	4	2

**Table 7 foods-12-03256-t007:** Kinetics of mangosteen vinegar fermentation with mixed cultures of *S. cerevisiae* TISTR 5279 and *G. oxydans* TRBC 4013.

Parameters	Values	Values
Fermentation time (d)	14	21
Acetic acid (%theoretical)	75.94 ± 1.99	85.23 ± 30
Y*ac/s* (g/g)	0.59	0.71
Q*ac* (g/L/h)	5.42	4.22
Residual sugars (g/L)	0.39 ± 0.10	0.49 ± 0.04
Initial viable cells (CFU/mL)		
*S. cerevisiae* TISTR 5279	7.5 × 10^7^	7.5 × 10^7^
*G. oxydans* TBRC 4013	5.5 × 10^7^	5.5 × 10^7^

Q*ac* = Acetic acid productivity (g/L/h), Y*ac/s* = Acetic acid yield (g/g). Mean ± SD from duplicate analysis.

**Table 8 foods-12-03256-t008:** Some properties of mangosteen vinegar.

Characteristics	Fermentation Times		
	Initial	Day 14	Day 21
pH	6.6 ± 0.00 b	2.95 ± 0.06 a	3.01 ± 0.11 a
Titratable acids (%)	0.60 ± 0.00 a	4.27 ± 0.33 b	4.94 ± 1.11 b
TFC (mg/100 mL)	20.38 ± 5.08 a	12.96 ± 0.65 a	11.63 ± 0.00 a
TPC (mgGAE/100 mL)	440.25 ± 2.59 a	359.67 ± 47.26 a	200.51 ± 7.08 b
DPPH radical inhibition (%)	18.75 ± 7.40 a	17.67 ± 0.22 a	15.17 ± 2.47 a
Turbidity	271.00 ± 1.41 a	19.11 ± 0.04 b	12.07 ± 0.99 c
Color values			
*L**	4.11 ± 0.01 a	3.62 ± 0.02 a	3.93 ± 0.66 a
*a**	−0.27 ± 0.01 a	−0.58 ± 0.02 a	−0.53 ± 0.25 a
*b**	0.34 ± 0.01 b	1.32 ± 0.01 a	1.35 ± 0.32 a

Mean values ± SD from triplicate analysis. Means values in the same row with different lowercase are significant differences (*p* < 0.05).

**Table 9 foods-12-03256-t009:** Mineral contents of mangosteen vinegar.

Minerals	Concentration (mg/100 mL)
Sodium	11.60 ± 0.05
Potassium	234.45 ± 1.06
Phosphorous	55.43 ± 3.33
Magnesium	96.02 ± 6.19
Calcium	34.84 ± 1.81
Sulphur	76.54 ± 41.95
Aluminium	nd
Manganese	nd
Iron	nd
Nickle	nd
Copper	nd
Zinc	nd
Boron	nd

Data are presented as means ± SD (*n* = 3). nd = not detected.

## Data Availability

The datasets generated for this study are available on request to the corresponding author.

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
