# Peer review of "Optimized Acetic Acid Production by Mixed Culture of Saccharomyces cerevisiae TISTR 5279 and Gluconobacter oxydans TBRC 4013 for Mangosteen Vinegar Fermentation Using Taguchi Design and Its Physicochemical Properties"

_foods, 2023, doi:10.3390/foods12173256_

Round 1

Reviewer 1 Report

The work presented deals with acetic acid production from an important fruit that is known for its human health boosting properties. However, the following issues should be considered:

1) The technical language needs extensive editing by a native English speaker

2) In the Introduction section, the author/s/ claimed that the bacterium culture used in this study can utilize five carbon sugar. However, the author/s/ failed to test for presence/absence of this sugar before and after the fermentation processes.

3) Methodology section needs improvement on statistical analysis and other issues indicated in the text

4) Some data in Table are not well presented and discussed (see text)

See text!

Author Response

Dear REviewers, I’m very appreciative of your comments and suggestion. All points raised by the reviewers were carefully addressed and answered point-by-point in the revised manuscript. A revision was made in highlighted red fonts for Reviewer1, green fonts for Reviewer2, light blue for Reviewer3 and yellow for authors. Some Figures1,3 were updated and all content was double-checked throughout the manuscript.  English was proofread by Asst. Professor. Dr. Budi Waluyo. If you have any questions, please do not hesitate to let me know.

Response to Reviewer 1 Comments

Point 1: The technical language needs extensive editing by a native English speaker

Response 1: It was proved.

Point 2: In the Introduction section, the author/s/ claimed that the bacterium culture used in this study can utilize five carbon sugar. However, the author/s/ failed to test for presence/absence of this sugar before and after the fermentation processes.

Response 2: Our preliminary study found that sugar composition in mangosteen juice is composed of sucrose, glucose, and fructose whereas xylose cannot be detected. This sugar is usually found in structural carbohydrates like hemicellulose So this experiment does not necessarily determine this sugar before and after fermentation.

Point 3: Methodology section needs improvement on statistical analysis and other issues indicated in the text

Response 3: It was revised.

Point 4: Some data in Table are not well presented and discussed (see text)

Response 4: It was rewritten as indicated in lines 425-429.

Reviewer 2 Report

The authors in the manuscript Optimized Acetic Acid Production by Mixed Culture of accharomyces cerevisiae TISTR 5279 and Gluconobacter oxydans TRBC 4013 for Mangosteen Vinegar Fermentation using Taguchi esign and Its Physicochemical Properties presented an optimized procedure for the production of mangosteen fruit vinegar. A properly edited manuscript, however, needs some minor additions.

1. line 92 give a more detailed description of the 5th stage of maturity

2nd row 94 how long the fruit was stored at 4 degrees Celsius, determination until analysis is not very accurate

3. For HPLC and ICP analyses, provide the methodology of sample preparation for analysis

4. In the discussion, expand the topic of comparing the obtained vinegars with vinegars obtained from a different matrix.

Author Response

Dear Reviewers,

          I’m very appreciative of your comments and suggestion. All points raised by the reviewers were carefully addressed and answered point-by-point in the revised manuscript. A revision was made in highlighted red fonts for Reviewer1, green fonts for Reviewer2, light blue for Reviewer3 and yellow for authors. Some Figures1,3 were updated and all content was double-checked throughout the manuscript. English was proofread by Asst. Professor. Dr. Budi Waluyo. If you have any questions, please do not hesitate to let me know.

Response to Reviewer 2 Comments

Point 1: line 92 give a more detailed description of the 5th stage of maturity

Response 1: It was revised as indicated in lines 97-98.

Point 2: row 94 how long the fruit was stored at 4 degrees Celsius, determination until analysis is not very accurate

Response 2: The pulp containing seed after peeling was stored at -200C, It was revised as indicated in line 98.

Point 3: For HPLC and ICP analyses, provide the methodology of sample preparation for analysis

Response 3: Methods were rechecked and revised as stated in lines 198-218.

Point 4: In the discussion, expand the topic of comparing the obtained vinegars with vinegars obtained from a different matrix.

Response 4: Additional discussion was performed as indicated in lines 373-378, 405-408.

Reviewer 3 Report

The manuscript presents laboratory production of mangosteen vinegar. The manuscript needs improvements:

- Glucononbacter oxydans is an acetic acid bacterium that does not resist acetic acid and ethanol. So, this species cannot be the strain used for this vinegar production. The strain needs identification, most appropriate by 16S rRNA gene comparative analysis.

- L46: acetic-containing: acetic acid-containing

- L173: what is the composition of PDA and GECA agar?

- Figure 1: it is not sharp

- L317: the conc. of EtOH is 3.95%, but the conc. of acetic acid is 75%; how is this possible?

- Figure 3: it is not possible to see the curves properly.

- L322: what is the difference between theoretical and predicted values?

- Manuscript needs English improvements.

- Some papers have been published on mangosteen vinegar production, which needs to be included in the discussion.

Author Response

Dear Reviewers,

        I’m very appreciative of your comments and suggestion. All points raised by the reviewers were carefully addressed and answered point-by-point in the revised manuscript. A revision was made in highlighted red fonts for Reviewer1, green fonts for Reviewer2, light blue for Reviewer3 and yellow for authors. Some Figures1,3 were updated and all content was double-checked throughout the manuscript. English was proofread by Asst. Professor. Dr. Budi Waluyo. If you have any questions, please do not hesitate to let me know.

Response to Reviewer 3 Comments

Point 1: Glucononbacter oxydans is an acetic acid bacterium that does not resist acetic acid and ethanol. So, this species cannot be the strain used for this vinegar production. The strain needs identification, most appropriate by 16S rRNA gene comparative analysis.

Response 1: -It is a very useful suggestion and thank you very much for your comments on this point.    

            However, the strain TBRC 4013 was purchased from the TBRC collection, it sources from vinegar.

   History: <---- BCC 14433 <---- NBRC, NBRC 3189 <---- IFO 3189 <---- NRRL B-147

Point 2: L46: acetic-containing: acetic acid-containing

Response 2: It was changed to “acetic acid-containing”, line 49.

Point 3: L173: what is the composition of PDA and GECA agar?

Response 3: It was revised in lines 110-111 for PDA and lines 117-118 for GECA

Point 4: Figure 1: it is not sharp

Response 4: Figure 1 was revised.

Point 5: L317: the conc. of EtOH is 3.95%, but the conc. of acetic acid is 75%; how is this possible?

Response 5: It was revised, 75 % is a 75 %theoretical, acetic acid was 5.34 ± 0.92%.

Point 6: Figure 3: it is not possible to see the curves properly.

Response 6: Figure 3 was revised.

Point 7: L322: what is the difference between theoretical and predicted values?

Response 7: The theoretical value of acetic acid is computed by converting carbohydrates based on sucrose concentration to acetic acid at a rate of 100%.

The predicted value is the technical term from statistical analysis of experimental

results performed by statistical design, Taguchi design was used in this study.

Point 8: - Manuscript needs English improvements.

Response 8: It was proved.

Point 9: - Some papers have been published on mangosteen vinegar production, which needs to be included in the discussion.

Response 9: It was stated in lines 406-409.

Round 2

Reviewer 1 Report

The author/s/ has/have incorporated my critical comments and he Ms has been improved.

Author Response

Dear Reviewers,

I’m very appreciative of your comments and suggestion.

Reviewer 3 Report

The identity of the bacterial strain has to be checked again. The Gluconobacter spp. cannot resist acetic acid and ethanol. They cannot be used for vinegar production. Although the strain has been bought from the culture collection, it has to be reidentified; mistakes can happen. However, the question remains of why the authors used Gluconobacter for vinegar production.
